# mTOR: Its Critical Role in Metabolic Diseases, Cancer, and the Aging Process

**DOI:** 10.3390/ijms25116141

**Published:** 2024-06-02

**Authors:** Sulaiman K. Marafie, Fahd Al-Mulla, Jehad Abubaker

**Affiliations:** 1Biochemistry and Molecular Biology Department, Dasman Diabetes Institute, P.O. Box 1180, Dasman 15462, Kuwait; 2Department of Translational Research, Dasman Diabetes Institute, P.O. Box 1180, Dasman 15462, Kuwait; fahd.almulla@dasmaninstitute.org

**Keywords:** mTOR, mTORC1, mTORC2, metabolic disease, cancer, aging, mTOR inhibitors

## Abstract

The mammalian target of rapamycin (mTOR) is a pivotal regulator, integrating diverse environmental signals to control fundamental cellular functions, such as protein synthesis, cell growth, survival, and apoptosis. Embedded in a complex network of signaling pathways, mTOR dysregulation is implicated in the onset and progression of a range of human diseases, including metabolic disorders such as diabetes and cardiovascular diseases, as well as various cancers. mTOR also has a notable role in aging. Given its extensive biological impact, mTOR signaling is a prime therapeutic target for addressing these complex conditions. The development of mTOR inhibitors has proven advantageous in numerous research domains. This review delves into the significance of mTOR signaling, highlighting the critical components of this intricate network that contribute to disease. Additionally, it addresses the latest findings on mTOR inhibitors and their clinical implications. The review also emphasizes the importance of developing more effective next-generation mTOR inhibitors with dual functions to efficiently target the mTOR pathways. A comprehensive understanding of mTOR signaling will enable the development of effective therapeutic strategies for managing diseases associated with mTOR dysregulation.

## 1. Introduction

The mammalian target of rapamycin (mTOR) is a protein kinase that plays an important role in regulating metabolism, cell growth and survival, immunity, and aging in response to different stimuli such as hormones, growth factors, nutrients, and stress signals [1,2]. Additionally, it acts as a nutrient sensor, playing important roles in various metabolic disorders such as diabetes, obesity, and cardiovascular diseases [3,4,5]. Dysregulation of mTOR signaling has also been implicated in many cancers where uncontrolled cell growth and division directly contribute to tumor development and progression [6,7]. More recently, studies have also revealed that rare mTOR gene variants were associated with severe COVID-19 outcomes [8]. mTOR is part of a vast network of signaling pathways that also play important roles in human diseases. It lies downstream of phosphatidylinositol 3 kinase (PI3K), where key players such as phosphoinositide-dependent kinase 1 (PDK1), protein kinase B (Akt), serum/glucocorticoid-regulated kinase 1 (SGK1), and AMP-activated protein kinase (AMPK) signaling contribute to overall mTOR functions [9,10]. mTOR exerts its effect by forming two distinct complexes; mTOR Complex 1 (mTORC1) and mTOR Complex 2 (mTORC2), each with unique roles and differential regulation. mTORC1 is composed of Raptor, telomere maintenance 2 (Tel2) and Tel2-interacting protein 1 (Tti1), mammalian lethal with SEC13 protein 8 (mLST8), proline-rich Akt substrate 40 kDa (PRAS40), and DEP domain-containing mTOR-interaction protein (DEPTOR) and is mainly triggered by growth factors and nutrients that regulate cell growth, survival, and autophagy. On the other hand, mTORC2 is composed of Rictor, Tel2 and Tti1, mLST8, mammalian stress-activated protein kinase-interacting protein 1 (mSin1), and DEPTOR. Like mTORC1, mTORC2 is also activated by growth factors but is less sensitive to nutrients. However, unlike mTORC1, mTORC2 is less understood but has been implicated to be involved in regulating the actin cytoskeleton as well as controlling cell size and glucose uptake [3,11,12,13].

Upon mTORC1 activation, it phosphorylates its downstream targets P70-S6K1 and 2 (S6K1 and S6K2) and 4E binding proteins 1 and 2 (4E-BP1 and 4E-BP2), whereas mTORC2 activation promotes Akt phosphorylation at serine 473 [14]. An essential nutrient-dependent interaction of mTORC1 involves Ras homolog enriched in brain (Rheb), which is crucial for phosphorylating its downstream targets and exerting its cellular functions [2]. In contrast to mTOR activation, rapamycin is one of the most well-known mTOR inhibitors and where mTOR gets its name from. Rapamycin, an FDA-approved drug, weakens mTORC1, causing the complex to dissociate and abolishing its cellular functions (Figure 1). Initially, it was believed that rapamycin exclusively targeted mTORC1. However, studies have shown that prolonged rapamycin treatment also impacts mTORC2 activity [15]. The clinical implications of mTOR inhibitors have been shown to be beneficial in many research areas. For instance, the treatment of certain types of cancer and other diseases characterized by abnormal and uncontrolled cell growth has benefited greatly from such inhibitors. Another mTOR inhibitor, everolimus, has been used in the treatment of kidney cancers, as well as breast and brain cancers [16,17,18]. Apart from cancer, mTOR inhibitors have also shown promising results in the treatment of genetic disorders such as tuberous sclerosis complex (TSC) and potential benefits in the treatment of neurodegenerative diseases such as Alzheimer’s disease [19].

Despite the differential regulation of both mTOR complexes, certain studies have demonstrated that they can crosstalk and mutually influence each other’s activity. One primary function of mTORC1 signaling is translation initiation, while mTORC2 signaling primarily regulates protein synthesis. A recent study by Oh et al. showed that both mTOR complexes crosstalk to achieve quality control during protein production [20]. Moreover, another study illustrated that dysregulation of mTORC1, either through rapamycin treatment or deletion of raptor, enhanced mTORC2 activity [21], whereas others implicated that S6K1 directly phosphorylates mTORC2 to dampen its activity [22].

Due to its numerous roles in human diseases, mTOR signaling is considered an important therapeutic target. Understanding its interactions with different cellular components offers valuable insights at both the cellular and disease levels. In this review, we will explore the importance of mTOR signaling in metabolic diseases and cancers, as well as the most recent findings on mTOR inhibitors and their clinical implications.

## 2. mTOR Signaling and Human Diseases

### 2.1. mTOR Regulation in Metabolic Diseases

Many studies have demonstrated the effect of mTOR dysregulation on metabolic diseases such as diabetes, obesity, and other cardiovascular diseases [23,24]. In diabetes, mTOR has been shown to play a key role in affecting insulin resistance and sensitivity, glucose uptake, lipid metabolism, and ketone production [23]. Diabetes is characterized by increased glucose and lipid levels that consequently disrupt the individual’s metabolic profile, leading to hyperglycemia, hyperlipidemia, and eventually insulin resistance [25,26]. One of the key targets of downstream mTORC1 signaling is insulin receptor substrate 1 (IRS-1) which prevents constitutive activation of the pathway [27]. IRS-1 hyperactivation due to continuous mTORC1 activity has been linked to both type 1 and type 2 diabetes where glucose fails to translocate to its surface receptor, resulting in increased glucose levels in the blood (Figure 2 bottom panel) [24]. Moreover, others have implicated a potential crosstalk between Akt and IRS-1 via mTORC2 under lipotoxic conditions in rat insulinoma cell lines. The study also demonstrated a reduction in glucose-stimulated insulin secretion (GSIS) under the same conditions [28]. A recent study by Brouwers et al. demonstrated a novel molecular mechanism of one of the most extensively studied proprotein convertases (PCs), furin, in regulating glucose levels in β-cells [29]. PCs are proteolytic enzymes that are ubiquitously involved in many biological processes and play crucial roles in regulating metabolic diseases [30,31]. In their study, they showed that the absence of furin-mediated mTORC1 activation led to β-cell dysfunction (Figure 2 bottom panel) [29]. The crucial roles of PCs in diabetes and its complications have been highlighted by others, emphasizing their importance for the development of future therapeutic interventions [32]. Moreover, mTORC1 hyperactivation has been shown to have a protective role in diabetes [33]. Ketogenesis involves the breakdown of fat in the liver into ketone bodies, which serve as an energy source for neighboring organs [34]. Uncontrolled ketogenesis causes increased levels of glucose, leading to hyperglycemia [35]. Ursino and colleagues demonstrate a protective role of mTORC1, where its increased activity in the liver hampers the production of ketone bodies. During fasting, the expression of the peroxisome proliferator-activated receptor (PPAR), the main transcriptional activator of ketone bodies, is increased. Inhibiting PPAR is essential to promote mTORC1 activity, and therefore, regulate ketogenesis [33]. As a result, mTORC1 has been considered a promising therapeutic target for restraining diabetic ketogenesis (Figure 2 top panel). Gayatri et al. demonstrated another example of an interplay between mTORC1 and mTORC2 under diabetic conditions where glutamine played a key role in regulating both pathways. Under conditions of low glutamine levels, mesenchymal stromal cells (MSCs) exhibit reduced mTORC1 activity, which in turn promotes mTORC2 stabilization of Runt-related transcription factor 2 (RUNX2), a critical transcription factor involved in bone differentiation within MSCs. They also showed that high glucose conditions trigger mTORC1 hyperactivity, suppressing mTORC2 in a glutamine-dependent manner. This highlights the important role of glutamine in controlling the nutrient switch between both mTORC1 and mTORC2 signaling, in addition to its importance in modulating the molecular cues involved in driving diabetes-induced bone adipogenesis [36].

Dysregulation of mTOR signaling has also been shown to play a role in the context of obesity. Elevated triglycerides (TGs) and a reduction in lipoprotein lipase (LPL) expression, two hallmarks of obesity, have been linked with the upregulation of sodium-coupled neutral amino acid transporter 2 (SNAT2) in a mTORC1-dependent manner [37]. Constitutive activation of mTORC1 has also been shown to exacerbate obesity due to the accumulation of hepatic lipid deposits, leading to insulin resistance [38,39]. mTOR signaling has also been associated with non-alcoholic fatty liver disease (NAFLD). Using a Mendelian randomization approach, single-nucleotide polymorphisms (SNPs) found in eukaryotic translation initiation factor 4E (eIF4E), a mTORC1 target, from Genome-Wide Association Studies (GWAS) demonstrated an effect on NAFLD. Increased plasma levels of eIF4E are associated with an increased risk of developing NAFLD. This emphasizes the importance of mTOR-related pathways in liver health (Figure 2 bottom panel) [6].

Other aspects of metabolic diseases have also been explored where mTOR has been shown to be central. In the context of neuronal energy consumption and synaptic plasticity, crosstalk between mTOR and its upstream kinase AMPK was established. Using computational modeling, the dynamics of both AMPK and mTOR activity were investigated in response to the amino acid glutamate. Moreover, the interplay between insulin receptors and calcium signaling on AMPK and mTOR activation was fed into the computational model to predict their potential effect on insulin signaling and the metabolic consumption rate, providing a better understanding of neuronal metabolism [1].

### 2.2. mTOR Signaling Promotes Cancer

The mTOR signaling pathway is exploited by approximately 30% of cancers, contributing to their development and progression due to its various roles that impact the cell cycle, growth, survival, and metabolism. It also regulates nutrient utilization and energy production, emphasizing its crucial metabolic role in promoting aspects of cancer progression and invasiveness such as angiogenesis and metastases [23,40]. Dysregulation of cancer-critical genes leads to mTOR hyperactivation, which in turn, increases the translation of pro-oncogenic proteins that directly affect cellular processes such as cell growth, migration, and de novo blood vessel formation [41]. For instance, the dysfunction of both mTORC1 targets eIF4E and 4E-BP1 has been implicated to affect such processes. Decreased 4E-BP1 and increased eIF4E levels directly contribute to mTORC1 hyperactivation, leading to pro-oncogenic traits [23,24,42]. Moreover, increased rates of protein synthesis are also accompanied by mTORC1 hyperactivation, further promoting cancer development [43]. In terms of tumor metabolism, mTOR’s role as a nutrient sensor is key due to its response to different nutrient cues such as glucose, amino acids, growth factors, and other stressors [23,44]. Controlling nutrient utilization and energy production provides the metabolic adaptations necessary for cancer cell survival and proliferation. Autophagy, a process that involves recycling cytoplasmic components in response to nutrient scarcity and lack of energy, is another cellular mechanism controlled by mTOR which suppresses carcinogenesis [45]. The absence of autophagy has been reported to promote the development of cancer [46,47]. mTORC1 phosphorylates one of autophagy’s key players, UNC-5-like autophagy-activating kinase 1 (ULK1), preventing it from complexing with other autophagic components, thereby activating autophagy [48]. mTORC2, on the other hand, has been shown to indirectly regulate mTORC1 and suppress autophagy (Figure 2 bottom panel) [20].

mTOR has been reported to be dysregulated in approximately 70% of breast cancers [49]. Triple-negative breast cancer (TNBC), a form of breast cancer that is more aggressive and has a higher susceptibility for metastases, has been reported to have mutations in the PI3K/Akt/mTOR pathway in 25% of cases [50,51]. Such disruption serves as a key TNBC survival and resistance-coping mechanism, making it a promising target for the treatment of TNBC. mTORC2 signaling also demonstrates cancer-promoting roles due to its regulation of Akt, glucose uptake, and apoptosis-promoting pro-proliferative cellular functions [23]. Rictor, one of the main components of mTORC2, has been reported to be crucial for the progression of human prostate cancer cell lines [52]. Phosphatase and tensin homolog (PTEN) negatively phosphorylates Rictor, leading to the dissociation of mTORC2 and consequently abolishing its activity [53]. This inhibitory mechanism is continuously exploited in both mouse and human cell line prostate cancer models where mTORC2 signaling is constitutively activated by PTEN deficiency or loss, contributing to prostate cancer progression (Figure 2 bottom panel) [52].

Despite mTOR hyperactivity in many cancers, some cytoplasmic regulators prevent mTOR activity, thereby preventing tumor growth. For instance, NLRC3 (nucleotide-binding domain and leucine-rich repeats 3) has been shown to negatively regulate mTOR signaling [19]. NLRs are a family of cytoplasmic sensors that regulate a range of biological functions involved in immunity against infectious diseases [54] and are central regulators of intestinal homeostasis [55]. Studies have shown that the expression of NLRC3 is dramatically reduced in patients with colorectal cancer (CRC), implicating its potential role in cancer development [56]. Karki and colleagues further confirm the role of NLRC3 in promoting colorectal tumorigenesis in mice in a mTOR-dependent manner. They demonstrate that upon NLRC3 binding to its receptor, it suppresses mTOR activation, preventing cellular proliferation by blocking the activation of Akt downstream of PI3K. They reveal a key mTOR inhibitory role for NLRC3 in mediating protection against CRC (Figure 2 top panel) [19]. Another research aspect that has been shown to play an important role in human diseases, including cancers, is the study of gut microbiota. p53-induced intestinal oncogenesis has been shown to be modulated epigenetically by the composition of the gut microbiome. Specifically, CRC was influenced by dietary proteins that impacted the gut microbiota, which is crucial for the development of effective cancer therapies [57,58]. A recent study compared the effect of two different diets, namely a casein protein diet (CTL) and a free amino acid (FAA)-based diet on the progression of CRC and gut microbiota in mice. They found that the FAA-based diet significantly attenuated CRC progression compared to the CTL diet, in addition to enriching beneficial gut bacteria. The attenuation was due to the downregulation of several cancer-associated pathways, including mTOR signaling [59]. This further emphasizes the vast role mTOR signaling plays in cancer metabolism and the importance of gut microbial composition in improving gut functions and preventing carcinogenic pathways.

## 3. mTOR Signaling and Life Extension

### 3.1. The Role of mTOR Signaling in Aging Is Evolutionarily Conserved

The role of mTOR signaling with regard to aging has been highlighted in different studies and is shown to be conserved from yeast to mammals [23,60]. Initial studies performed on *C. elegans* demonstrated the role of mTOR and Raptor orthologs on the longevity and the extension of life span in worms. They showed that decreased expression in both mTOR and Raptor orthologs had a direct effect on life span extension [61,62]. Similar studies performed on other organisms such as budding yeast [63], *Drosophila* [64], and mouse models resulted in consistent findings [65,66]. Nutrient deprivation was suggested as a means of lowering the organism’s metabolic rate, leading to life span extension. Indeed, limiting nutrients led to a reduction in mRNA translation driven by mTORC1, eventually lowering oxidative stress and preventing the accumulation of toxic metabolites, causing an increase in life expectancy [23]. Others reported similar findings in mammals, further confirming the link between lower mTORC1/S6K1 activity and increased life span (Figure 2 top panel) [67]. Additionally, the mTOR inhibitor rapamycin has also been shown to have a role in life span extension across different organisms, making it the only known pharmaceutical drug to directly regulate aging [68,69,70,71].

### 3.2. Autophagy Promotes Life Span Extension in a mTOR-Dependent Manner

One explanation that correlates mTORC1 activity to life span extension is the role of autophagy in mTOR signaling. Inhibition of mTORC1 promotes autophagy, which in turn cleanses unwanted cytosolic proteins and reduces the accumulation of toxic metabolites, leading to a reduction in cellular stress and an extension of life span [23]. To date, it has been established that mTOR signaling inhibits autophagy, and autophagy declines with aging [72,73]. This has been further highlighted in recent studies investigating senescent cells in the context of aging. One of the hallmarks of aging is the accumulation of senescent cells which are commonly present in many age-related diseases and cancers [74]. Recently, studies have demonstrated increased expression of programmed death-ligand 1 (PD-L1) protein in senescent cells. Upregulation of PD-L1 protects senescent cells from clearance by the PD-1 checkpoint receptor, and mTOR signaling has been shown to be one of the key stimulators of PD-L1 [75]. Thus, the PD-1/PD-L1 checkpoint is crucial for promoting the accumulation of senescent cells and slower aging (Figure 2 top panel). Moreover, there has been evidence demonstrating that mTOR-mediated inhibition of autophagy stimulates PD-L1 expression [76,77,78]. Others also showed similar findings where autophagy inhibition, by either pharmacological inhibitors or knockout of autophagic genes, increased PD-L1 expression in mouse and human cancer cell models. It is known that PD-L1 possesses functions other than cell-intrinsic ones [79,80]. One study demonstrated that PD-L1 acts as an upstream regulator of mTOR, activating the Akt–mTOR axis and promoting the proliferation of human ovarian cancer cells [81]. Interestingly, others have reported PD-L1’s indirect mediation of mTOR signaling by directly preventing the autophagy flux in tumor cells [82]. This highlights the bidirectional role PD-L1 plays in regulating mTOR and autophagy, something that seems to be cell-specific in certain cases. Taken together, these findings suggest that PDL-1 is a potential candidate for anti-aging therapies and interventions.

### 3.3. Mitochondrial Proteins Regulate Life Span via mTOR Signaling

Another hallmark of aging is the process of controlling the coordination between mitochondrial and nuclear activities that are essential for overall cellular respiration. Disruption of such coordination leads to mitochondrial dysfunction, affecting overall aging [74,83]. Signaling pathways drive the crosstalk between the mitochondria and the nucleus, which in turn affects nuclear gene expression to maintain mitochondrial homeostasis [84,85]. Clk-1 is one of the key proteins that has been shown to play a role in such communication, affecting cellular respiration and longevity [86]. A study conducted by Monaghan et al. uncovered a direct nuclear role for Clk-1 in regulating life span that is conserved from *C. elegans* to humans [87]. It was later revealed, for the first time, that Clk-1 could be regulated by mTOR via the AMPK–mTOR axis [88]. Thus, these studies emphasize the crucial role mTOR signaling plays in regulating life span by communicating and linking with different biological processes by sensing various cellular cues.

## 4. Inhibitors of mTOR Signaling Pathways

### 4.1. First-Generation mTOR Inhibitors

So far, we have shown the central role mTOR signaling plays in regulating various biological processes, from metabolism to cell growth and survival, in response to a range of cellular stimuli. We have also reported how its dysregulation is key in promoting a variety of human diseases, ranging from metabolic diseases to cancer. Therefore, mTOR inhibitors have emerged as a significant therapeutic strategy for such diseases. We mentioned earlier in this review the role of rapamycin in regulating mTOR signaling. Rapamycin, also known as sirolimus, is considered a first-generation allosteric mTOR inhibitor and has been approved for the treatment of several types of cancers [89,90]. Its mechanism of action is dependent on it forming a complex with FK506-binding protein 12 (FKBP12), which in turn binds to the FKBP-rapamycin-binding (FRB) domain of mTOR to exert its inhibitory effect (Figure 3) [11,23,91,92]. Nevertheless, the first-generation inhibitors partially inhibit mTOR signaling due to their allosteric nature. They partially inhibit mTORC1 activity while cells retain mTORC2 activity due to the feedback activation of PI3K/Akt signaling [93]. This was highlighted in a clinical trial conducted by Cloughesy et al., where a number of glioblastoma (GBM) patients deficient in PTEN were treated with rapamycin and showed a decrease in tumor proliferation. However, the remaining patients displayed increased Akt activation levels, implicating the failure of rapamycin to consistently treat individuals suffering from GBM [94].

Rapalogs (rapamycin analogs) are another set of first-generation inhibitors that act similarly on mTOR and have been implicated in various clinical trials. In a phase 1 trial also looking at patients with GBM, individuals were treated with the rapalog everolimus in conjugation with the standard treatment of temozolomide (TMZ) and radiation therapy (RT). This also tested the efficacy of everolimus conjugation therapy compared to the standard treatment alone. Out of the 18 patients diagnosed with GBM, nine individuals developed advanced grades of toxicities, compared to only four who partially responded to everolimus [95]. Another study with a larger cohort of 100 newly diagnosed GBM patients showed similar results, implicating that using conjugation therapy with everolimus has no survival benefits compared to the standard therapy of TMZ and RT alone [96]. In a dose-dependent trial, temsirolimus’ efficacy was also investigated in GBM. In the first cohort of the study, temsirolimus in conjugation with RT alone and not TMZ reduced the infection of a few patients with the aid of prophylactic antibiotics. However, the second cohort displayed contrasting results, where worsening of pre-existing infections occurred [97]. Despite being widely used, first-generation mTOR inhibitors’ limited effect on patients’ disease progression demands the development of more robust inhibitors that provide complete mTOR inhibition.

### 4.2. Second-Generation mTOR Inhibitors

The main aim of active site inhibitors is to simultaneously target mTOR signaling and its feedback loops, which was lacking from the first-generation inhibitors. Consequently, second-generation mTOR inhibitors have been developed (Figure 3). They are ATP-competitive dual inhibitors that target both mTORC1 and mTORC2 and have shown promising results, providing complete inhibition of the mTOR signaling pathway [98,99].

Sapanisertib, previously known as TAK-228 or MLN0128, is a potent dual inhibitor of mTORC1 and TORC2 that recently demonstrated promising anti-tumor effects in endometrial and renal cell carcinomas [100,101,102,103,104]. Voss et al. showed that sapanisertib exerts its anti-tumor effects by regulating the downstream targets of both mTORC1 (S6K1 and p4EBP1) and mTORC2 (PRAS40) [104]. Another study conducted by Coleman et al. also demonstrated anti-tumor activity using sapanisertib. However, they used sapanisertib treatment in combination with ziv-aflibercept across tumor types [105]. Ziv-aflibercept is a recombinant protein that consists of human vascular endothelial growth factor (VEGF) receptor extracellular domains fused to human immunoglobulin G1. VEGF is key for the angiogenic activity of tumors and studies have shown that it is downregulated by sapanisertib, decreasing angiogenic activity [106]. The combination of sapanisertib with ziv-aflibercept in treating patients with solid tumors showed promising clinical benefits, where 80% of patients achieved disease control. Moreover, a few patients demonstrated mutations in key players involved in mTOR hyperactivity, namely TSC1 and Akt, suggesting the important role of combination therapy for patients with advanced solid tumors [105]. Sapanisertib has also been used in conjugation with metformin, a widely used drug for the treatment of type 2 diabetes [107], in a phase 1 trial [108]. Metformin also exhibits mTOR inhibitory functions [109,110], where the combined therapy could potentially enhance the anti-tumor effects of sapanisertib. A study by Subbiah and colleagues reported that patients with solid tumors, exhibiting advanced metastases, who were resistant to standard treatments received sapanisertib treatment with metformin in a dose-dependent manner for 14 days. Their study concluded that the safety, tolerability, and efficacy of the combination therapy was comparable to the standard therapy for the treatment of advanced solid tumors [108].

Several second-generation ATP-competitive inhibitors of mTOR have been identified, including PP242, KU0063794, AZD3147, and eCF309. Pyrazolopyrimidines, such as PP242 and PP30, have been shown to be more selective towards mTOR inhibition in relation to PI3K and other kinases. Additionally, KU0063794 displayed promising results in suppressing cell cycle/proliferation compared to the well-established PI3K inhibitor, LY294002 [111]. The second generation of mTOR inhibitors has shown promising results in preclinical and clinical trials by providing more complete inhibition of the mTOR signaling pathway. This allowed others to explore different approaches to identify additional ATP-competitive inhibitors for mTOR, such as using virtual docking, theoretical biology tools, and molecular modeling [112], emphasizing the importance of regulating mTOR signaling in human disease.

### 4.3. Third-Generation mTOR Inhibitors

Recently, a third-generation inhibitor of mTOR has emerged that targets multiple domains in both mTOR complexes (Figure 3). RapaLink-1 is an example of this new generation of inhibitors that resembles rapamycin in structure and links to mTOR, resulting in potent inhibition of mTORC1. Despite the unknown immunosuppressive potential of the drug, Wang and colleagues investigated its therapeutic efficacy in organ transplantation. They demonstrated that RapaLink-1 outperformed rapamycin’s inhibitory role in T-cell proliferation and drastically prolonged graft survival time. Furthermore, the reduction in graft rejection was associated with reduced mTORC1 activity. This study was the first to demonstrate the effectiveness of third-generation mTOR inhibitors in the context of organ transplantation [113]. Further studies on these third-generation inhibitors are important to emphasize their potential for the mitigation of metabolic diseases and cancers.

**Figure 3 ijms-25-06141-f003:**
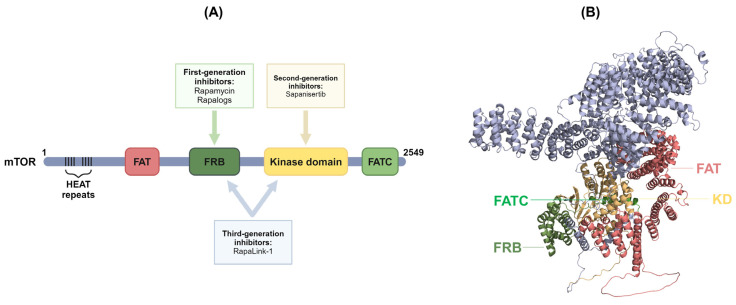
(**A**) An illustration of the different generations of mTOR inhibitors and their corresponding target domains on the mTOR molecule. First-generation inhibitors are shown to target the FRB domain of the mTOR protein, whereas second-generation inhibitors target the active site of the kinase domain. Third-generation inhibitors possess dual functions in targeting both the FRB and kinase domains of mTOR. (**B**) A 3-D structural representation of the mTOR molecule generated using Alpha Fold 2.0 [114], indicating the four mTOR domains. mTOR, mammalian target of rapamycin; HEAT, Huntingtin, elongation factor 3, PP2A and TOR1; FAT, FRAP, ATM, and TRAP; FRB, FKBP-rapamycin-binding; KD, kinase domain; FATC, C-terminal FAT (Created with BioRender.com accessed on 25 May 2024).

### 4.4. Other Inhibitors of the mTOR Signaling Pathway

As mentioned earlier, metformin has been shown to have a dual role in the treatment of type 2 diabetes and cancers by regulating mTOR signaling [107]. The primary mode of action metformin exerts on mTOR is the alteration of cellular energy metabolism, which in turn stimulates AMPK, attenuating Rheb and affecting the mTORC1 signaling axis. Additionally, cell stress-mediated activation of AMPK further reduces mTORC1 activity by suppressing Rag GTPases (key activators of mTORC1 in response to amino acids flux) [109,110]. Moreover, Gayatri et al. reported that the activation of AMPK by metformin also inhibits high glutamine-induced mTORC1 hyperactivation via the mTORC2–Akt axis, solidifying the importance of the crosstalk between both mTORCs mentioned earlier in this report [36]. However, recent studies have shown that metformin can inhibit mTORC1 independently of AMPK [109,110]. Others have also demonstrated the AMPK-independent inhibitory role of metformin in mTOC1 signaling. Metformin was shown to induce the expression of activating transcription factor 4 (ATF4), regulated in development and DNA damage-response 1 (REDD1), and Sestrin2, all of which are important for the attenuation of mTORC1 [109]. The AMPK-dependent and independent regulatory roles of metformin in mTOR signaling are crucial for the overall regulation of cellular energy metabolism, and therefore for the treatment of various metabolic diseases.

A study by Lin et al. demonstrated the use of the amino sulfonic acid Taurine (Tau) to limit the invasion and metastasis of TNBC in a mTOR-dependent manner. The effects of Tau on macrophage polarization were investigated in relation to cell growth, invasion, and metastasis both in vivo and in vitro [115]. Macrophages are polarized into two main states; M1 and M2. M1 macrophages are regarded as anti-tumor, whereas M2 (also known as tumor-associated macrophages, “TAMs”) are known to promote tumor progression [116,117]. In their study, they used TNBC cell lines to investigate the effect of Tau on tumor growth and invasion. Tau inhibited breast cancer metastasis in vivo and prevented cell invasion in vitro by altering the polarization of the M2 macrophages. In addition, Tau upregulated PTEN expression that consequently suppressed PI3K/Akt signaling. This upregulation further promoted M1 polarization of macrophages, ultimately inhibiting the metastasis of TNBC cells. These observations could act as a potential therapeutic approach to influence and control cancer progression and metastasis [115].

## 5. Conclusions

mTOR is a central physiological regulator that integrates signals from different environmental cues to regulate fundamental cellular processes. These range from protein synthesis to cell growth, survival, and death. We highlighted, in this review, the dysregulation of mTOR signaling and its strong association with several diseases such as metabolic diseases and cancers, as well as its role in life span regulation. Fully understanding mTOR signaling is important and is considered a promising target for therapeutic intervention, as indicated in the aforementioned studies (Figure 4). Future directions toward further exploring mTOR signaling are key, as indicated by the continuous development of more effective and efficient next-generation mTOR inhibitors. This includes developing drugs that serve as both allosteric and ATP-competitive inhibitors of mTOR. Exploring more combination therapies has also proven to be successful and is important, given their therapeutic efficacy in clinical cancer trials. Identifying predictive biomarkers such as the phosphorylation status and/or the presence of mutations in key effectors of mTOR signaling (e.g., PI3K and Akt) could help predict responses to PI3K/mTOR inhibitors. Such research advances our understanding of mTOR signaling to develop more effective therapeutic strategies for the management of diseases associated with mTOR dysregulation.

## Figures and Tables

**Figure 1 ijms-25-06141-f001:**
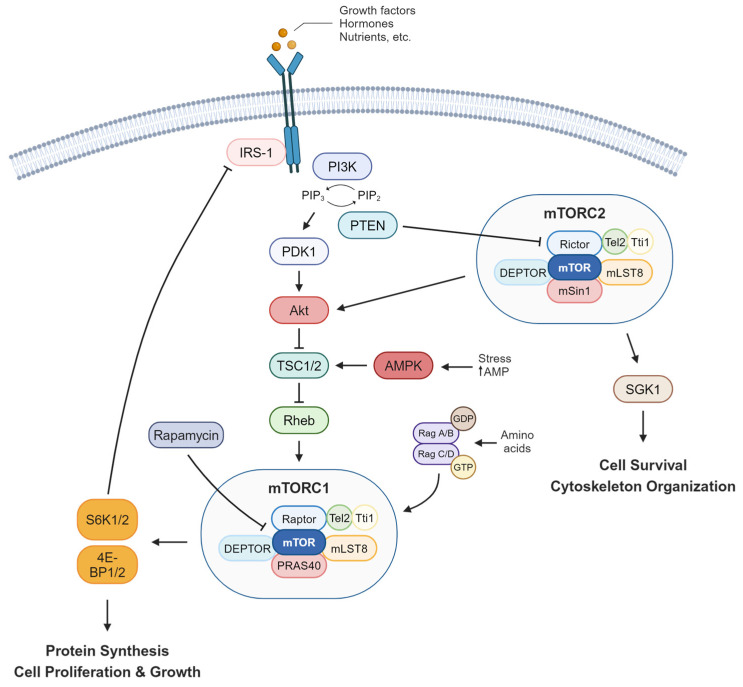
A schematic representation of the mTOR signaling pathway and the cellular functions driven by the formation of its two main complexes, mTORC1 and mTORC2. mTORC1 is primarily activated by growth factors and nutrients, which in turn triggers the activation of the PI3K/PDK1/Akt cascade. Akt promotes mTORC1 activation by inhibiting the TSC1/2 dimer, allowing Rheb-mediated activation of mTORC1. Both S6K1/2 and 4E-BP1/2 are the main downstream targets of mTORC1 signaling which play crucial roles in protein synthesis and consequently, promote cell proliferation and growth. S6K1 prevents continuous activation of mTORC1 signaling by phosphorylating IRS-1 via its negative feedback loop. Rag GTPases also promote mTORC1 activity in the presence of mitogens such as amino acids. Conversely, under stress conditions or when AMP levels increase, AMPK negatively affects mTORC1 activity by promoting the inhibitory effects of TSC1/2 on mTORC1. Additionally, rapamycin also acts as a negative regulator of mTORC1 by directly inhibiting the mTOR protein itself. Upon mTORC2 activation, it phosphorylates its downstream targets Akt and SGK1, promoting cell survival and cytoskeleton organization. PTEN negatively regulates mTORC2 activity by targeting one of its main components, Rictor, leading to the dissociation of the complex. mTOR, mammalian target of rapamycin; mTORC1 and 2, mTOR complex 1 and 2; PI3K, phosphatidylinositol 3 kinase; PIP2, phosphatidylinositol 4,5-bisphosphate; PIP3, phosphatidylinositol 3,4,5-trisphosphate; PDK1, phosphoinositide-dependent kinase 1; Akt, protein kinase B; TSC1/2, tuberous sclerosis proteins 1 and 2; Rheb, Ras homolog enriched in brain; AMPK, AMP-activated protein kinase; S6K1/2, P70-S6K1 and 2; 4E-BP1/2, 4E binding proteins 1 and 2; IRS-1, insulin receptor substrate 1; PTEN, phosphatase and tensin homolog; SGK1, serum/glucocorticoid-regulated kinase 1; Tel2, telomere maintenance 2; Tti1, Tel2-interacting protein 1 (Tti1), mLST8, mammalian lethal with SEC13 protein 8; PRAS40, proline-rich Akt substrate 40 kDa; DEPTOR, DEP domain-containing mTOR-interaction protein; mSin1, mammalian stress-activated protein kinase-interacting protein 1 (Created with BioRender.com accessed on 25 May 2024).

**Figure 2 ijms-25-06141-f002:**
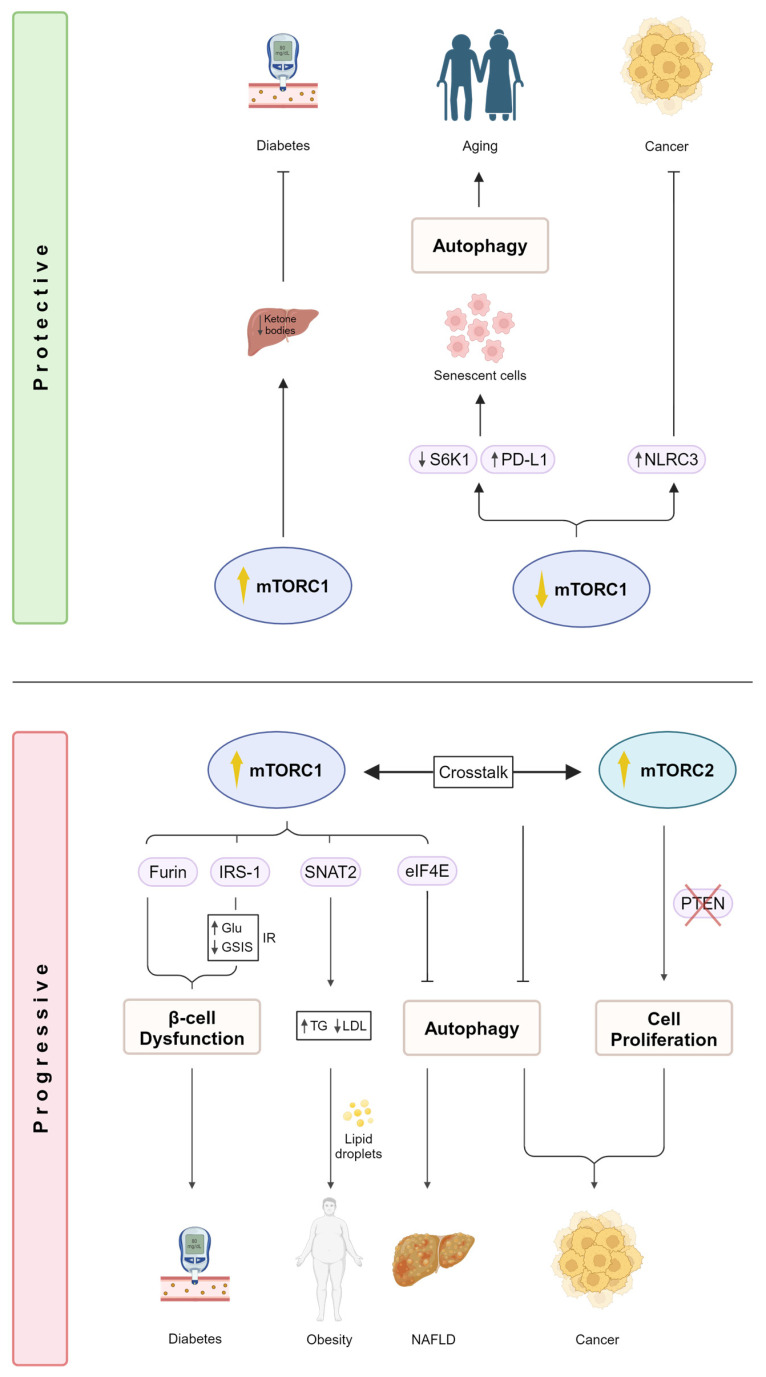
The roles of mTORC1 and mTORC2 in regulating metabolic diseases, cancer, and aging. Differential mTORC1 activity has been shown to be protective against diabetes and cancer, in addition to prolonging life span (**top panel**). mTORC1 hyperactivity hampers the production of ketone bodies in the liver, protecting against diabetic ketogenesis. In contrast, a reduction in mTORC1 activity promotes the accumulation of senescent cells due to the downregulation of S6K1 and upregulation of PD-L1, which in turn triggers autophagy, slowing the aging process. Moreover, increased levels of NLRC3 due to reduced mTORC1 signaling exert inhibitory effects against cancer, further improving overall health. However, hyperactivation of both mTORC1 and mTORC2 induces progressive outcomes towards metabolic diseases and cancers (**bottom panel**). Increased activities of both furin and IRS-1 due to continuous mTORC1 activation cause high glucose levels, reduced GSIS, and promote insulin resistance. Taken together, such disruptions directly contribute to β-cell dysfunction, and eventually, diabetes. Increased TGs and lower LDL promote the accumulation of lipid droplets, leading to obesity in a mTORC1-SNAT2-dependent manner. Inhibiting autophagy also plays a role in exacerbating metabolic outcomes that lead to NAFLD driven by eIF4E. Moreover, the crosstalk between both mTOR complexes has also been shown to further inhibit the autophagic pathway, leading to cancer. mTORC2 hyperactivation, due to PTEN loss, leads to continuous cell proliferation, further driving cancer progression. mTORC1 and 2, mammalian target of rapamycin complex 1 and 2; S6K1, P70-S6K1; PD-L1, programmed death-ligand 1; NLRC3, nucleotide-binding domain and leucine-rich repeats 3; IRS-1, insulin receptor substrate 1; SNAT2, sodium-coupled neutral amino acid transporter 2; eIF4E, eukaryotic translation initiation factor 4E; PTEN, phosphatase and tensin homolog; Glu, glucose; GSIS; glucose-stimulated insulin secretion; IR, insulin resistance; TG, triglyceride; LDL, lipoprotein lipase; NAFLD, nonalcoholic fatty liver disease (Created with BioRender.com accessed on 25 May 2024).

**Figure 4 ijms-25-06141-f004:**
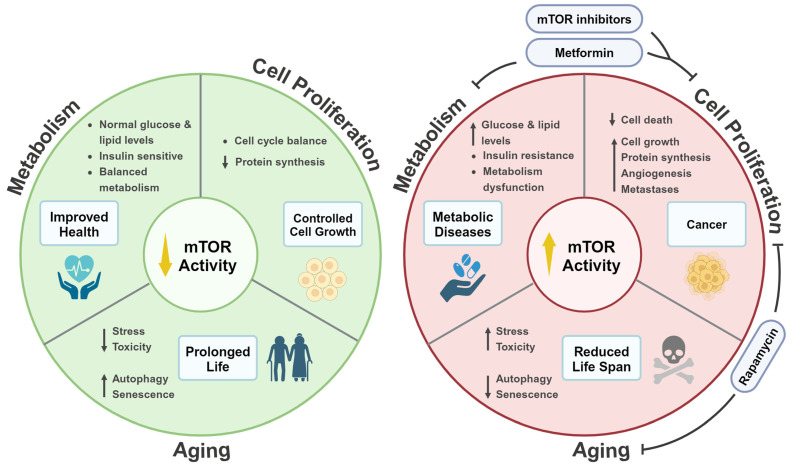
A summary of the central role mTOR signaling plays in regulating various cellular processes and its association with metabolic diseases, cancers, and aging. The left panel (green) illustrates the physiological state where mTOR signaling is reduced. Under these conditions, cells are characterized by having a balanced metabolism (i.e., normal glucose and lipid levels), being sensitive to insulin, a reduction in stress and toxicity, and having a balanced cell cycle. As a result, lower mTOR activity leads to an improvement in overall health, controlled cell growth, and eventually, a prolonged life span. Upon mTOR hyperactivity (right panel, red), glucose and lipid levels increase, cells undergo insulin resistance, stress and toxicity levels increase, and the cell cycle is disrupted, promoting uncontrolled cell growth. Such disruptions in mTOR signaling directly contribute to the progression of metabolic diseases (e.g., diabetes and obesity) and cancers, consequently reducing life span. The right panel also shows the impact of mTOR inhibitors, such as metformin and rapamycin, and their effects in mitigating disease progression and life span extension by reversing the drastic effects caused by mTOR hyperactivity (Created with BioRender.com accessed on 25 May 2024).

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
