# Peer review of "mTOR: Its Critical Role in Metabolic Diseases, Cancer, and the Aging Process"

_ijms, 2024, doi:10.3390/ijms25116141_

Round 1

Reviewer 1 Report

Comments and Suggestions for Authors

Your article emphasizes the significance of mTOR signaling, and highlights the critical components of intricate network which contributes to disease. However, there are deficiencies in language expression.So I think this article could be revised by following suggestions.

Comments on the Quality of English Language

1. Line 22: Change pathway to pathways .

2. Line 62: Delete to be .

3. Line 67: Delete have shown.

4. Line 76: Change  dampening  to  to dampen  .

5. Line 77: Change  role  to  roles  . 

6. Line 77: Add as  behind  is considered .

7. Line 104: Change for  to  of .

8. Line 111: Change serve to  serves.

9. Line 180: Add  ,  in front of  implicating .

10. Line 182: Change  an  to  a .

11. Line 187: Change  that  to  which .

12. Line 194: Modify CRC can be influenced by ... the effectiveness of cancer therapy. to make it clearer.

13. Line 239: What does  this  mean? Please make it clearer.

14. Line 358: Change  an  to  a .

15. Line 395:Change  that  to  which .

Author Response

Authors’ response to Reviewer 1:

We thank the reviewer for the constructive feedback and input. The points raised were helpful and strengthened the impact of the review. We addressed the points raised by the reviewer and modified the manuscript accordingly by highlighting the changes within the manuscript in yellow. We hope the adjusted manuscript will satisfy the key comments pointed out by the reviewer. 

Reviewer 1:

Your article emphasizes the significance of mTOR signaling, and highlights the critical components of intricate network which contributes to disease. However, there are deficiencies in language expression. So I think this article could be revised by following suggestions.

Comments on the Quality of English Language

Response: We thank the reviewer for the comments and feedback. The language has been adjusted as the reviewer suggested for Lines 22, 63, 69, 103, 104, 134, 216, 218, 442, 463 which have been highlighted in yellow within the manuscript.

Line 194: Modify “CRC can be influenced by ... the effectiveness of cancer therapy.” to make it clearer.

Response: Thank you for pointing this out and we agree that it needs further clarification. We have addressed this point and clarified it as follows:

Line 223-226:

“p53-induced intestinal oncogenesis has been shown to be modulated epigenetically by the gut microbiome composition. Specifically, CRC was influenced by dietary proteins that impacted the gut microbiota which is crucial for the development of effective cancer therapies”.

Line 239: What does “ this ” mean? Please make it clearer.

Response: We thank the reviewer for pointing this out. We agree that the sentence wasn’t clear, and we decided to delete the whole sentence. It did not add value to the paragraph and sounded redundant and confusing.

Reviewer 2 Report

Comments and Suggestions for Authors

This is very well written review; the information is well synthesized. However, there is room for improvement. Please see the followings:

Figure 2 is confusing; a better one is needed. It is not cited in proper context. It should be somewhere in the introduction.

mTOR is discussed in first seven pages; its components and interactions with different signaling pathways are discussed without any visual aid. Multiple figures depicting these interactions should accompany this section.

It is not very clear how different inhibitors interact with mTOR and its components and inhibit mTOR.

Line 182: NLRC3 itself is a receptor; mention its ligands.

Comments on the Quality of English Language

None

Author Response

Authors’ response to Reviewer 2:

We thank the reviewer for the constructive feedback and input. The points raised were helpful and strengthened the impact of the review. We addressed the points raised by the reviewer and modified the manuscript accordingly by highlighting the changes within the manuscript in yellow. We hope the adjusted manuscript will satisfy the key comments pointed out by the reviewer. 

Reviewer 2:

This is very well written review; the information is well synthesized. However, there is room for improvement. Please see the followings:

  1. Figure 2 is confusing; a better one is needed. It is not cited in proper context. It should be somewhere in the introduction.

Response: We thank the reviewer for their comments and highlighting this point. The reason for having it at the very end of the manuscript is that it acts as a summary of the findings mentioned in the different studies of this review. It also illustrates some of the key the mTOR inhibitors mentioned in Chapter 4 that mitigates the effects of mTOR hyperactivity. For this reason, we did not include it in the Introduction section. However, we did include two additional figures (Figures 1 and 2) within the manuscript, and one of them has been added in the introduction as the reviewer suggested to further clarify mTOR signaling along with the roles of both mTORC1 and mTORC2 in regulating different cellular processes (Figure 1). This can be found in line 71.

We also agree with the reviewer that Figure 4 (formerly Figure 2) was not discussed clearly, and we addressed this point by expanding the figure legend and discuss it in more detail as shown below:

Line 472-484:

“Figure 4. A summary of the central role mTOR signaling plays in regulating various cellular processes and its association with metabolic diseases, cancers, and aging. The left panel (green) illustrates the physiological state where mTOR signaling is reduced. Under these conditions, cells are characterized by having a balanced metabolism (i.e. normal glucose and lipid levels), being sensitive to insulin, a reduction in stress and toxicity, and having a balanced cell cycle. As a result, lower mTOR activity leads to an improvement in overall health, controlled cell growth, and eventually, a prolonged life span. Upon mTOR hyperactivity (right panel, red), glucose and lipid levels increase, cells undergo insulin resistance, stress and toxicity levels in-crease, and the cell cycle is disrupted promoting uncontrolled cell growth. Such disruptions in mTOR signaling directly contribute to the progression of metabolic diseases (e.g. diabetes and obesity), cancers, and consequently, reducing life span. The right panel also shows the impact of mTOR inhibitors, like metformin and rapamycin, and their effects in mitigating diseases progression and life span extension by reversing the drastic effects caused by mTOR hyperactivity (Created with BioRender.com).”

  1. mTOR is discussed in first seven pages; its components and interactions with different signaling pathways are discussed without any visual aid. Multiple figures depicting these interactions should accompany this section.

Response: We thank the reviewer for the constructive feedback and for pointing this out. We agree that the manuscript felt very lengthy and lacked any visual aid. We addressed this point by including two additional figures, Figure 1 and Figure 2. Figure 1 as part of the introduction illustrates the two main complexes of the mTOR signaling pathway along with their components and their different cellular roles. Figure 2 is part of Chapter 2 that discusses the roles of mTORC1 and mTORC2 in regulating metabolic diseases (i.e. diabetes, obesity, and NAFLD) and cancers, as well as their roles in aging. Figure 1 can be found in Line 71, while Figure 2 in Line 284.

  1. It is not very clear how different inhibitors interact with mTOR and its components and inhibit mTOR.

Response: We thank the reviewer for their feedback and for raising this point. We have mentioned in the manuscript the different modes of action each generation of inhibitors exert on mTOR and how they contribute to abolishing its cellular functions. Regarding the first-generation inhibitors (e.g. rapamycin), we indicated that they act on the FRB domain of mTOR which causes the dissociation on mTORC1 as a whole abolishing its activity. Below are the lines within the manuscript:

Line 59-62:

“Rapamycin, an FDA-approved drug, weakens mTORC1 causing the complex to dissociate abolishing its cellular functions (Figure 1). Initially, it was believed that rapamycin exclusively targeted mTORC1. However, studies have shown that prolonged rapamycin treatment also impacts mTORC2 activity [15]”

Line 332-338:

“Its mechanism of action is dependent on it forming a complex with the FK506-binding protein 12 (FKBP12), which in turn, binds to the FKBP-rapamycin-binding (FRB) domain of mTOR exerting its inhibitory effect (Figure 3) [11, 23, 92, 93]. Nevertheless, the first-generation inhibitors partially inhibit mTOR signaling due to their allosteric nature. They partially inhibit mTORC1 activity while cells retain mTORC2 activity due to the feedback activation of PI3K/Akt signaling [94]”

For the second-generation inhibitors, they act on the active site of mTOR’s kinase domain rather than the FRB domain, which in turn leads to the dissociation of both mTOR complexes. The dual inhibitory effect on both mTORC1 and mTORC2 serves as a more potent inhibitor compared to the first-generation inhibitors. Below are the lines where we address this:

Line 361-365:

“The main aim of active site inhibitors is to simultaneously target mTOR signaling and its feedback loops, which was lacking from the first-generation inhibitors. Consequently, second-generation mTOR inhibitors have been developed (Figure 3). They are ATP-competitive dual inhibitors that target both mTORC1 and mTORC2 and have shown promising results providing complete inhibition of the mTOR signaling pathway [99, 100]”

Unlike the first two generation of inhibitors, Metformin does not directly interact with mTOR to exerts its inhibitory functions. Instead, it acts indirectly on mTOR in AMPK-dependent and independent manners. Similarly, Tau also indirectly affect mTOR signaling by regulating PTEN, which in turn, prevents mTOR signaling. This can be seen in the lines below:

Line 425-438:

“The primary mode of action metformin exerts on mTOR is the alteration of cellular energy metabolism, which in turn stimulates AMPK attenuating Rheb affecting the mTORC1 signaling axis. Additionally, cell stress-mediated activation of AMPK further reduces mTORC1 activity by suppressing Rag GTPases (key activators of mTORC1 in response to amino acids flux) [110, 111]. Moreover, Gayatri et al. reported that the activation of AMPK by metformin also inhibits high glutamine-induced mTORC1 hyperactivation via the mTORC2-Akt axis solidifying the importance of the crosstalk between both mTORCs mentioned earlier in this report [36]. However, recent studies have shown that metformin can inhibit mTORC1 independent of AMPK [110, 111]. Others have also demonstrated the AMPK-independent inhibitory role of metformin on mTOC1 signaling. Metformin was shown to induce the expression of activating transcription factor 4 (ATF4), regulated in development and DNA damage-response 1 (REDD1), and Sestrin2, all of which are important for the attenuation of mTORC1 [110]”

Line 449-452:

“In addition, Tau upregulated PTEN expression that consequently suppressed PI3K/Akt signaling. This upregulation further promoted M1 polarization of macrophages, and ultimately, inhibited the metastasis of TNBC cells.”

For the third-generation inhibitors, they posses dual roles in targeting both the FRB and kinase domains of mTOR. Targeting both domains ensures the complete dissociation of both mTORCs as well as ensuring the complete abolishment of mTOR activity. We have addressed this below:

Line 402-405:

“Recently, a third-generation inhibitor of mTOR has emerged which targets multiple domains in both mTOR complexes (Figure 3). RapaLink-1 is an example of this new generation of inhibitors that resemble rapamycin in structure and links to mTOR, resulting in potent inhibition of mTORC1.”

However, despite mentioning the different modes of action of each class of inhibitor, we agree with the reviewer that further clarification is needed. We added an additional panel to Figure 3 (panel B) which shows the 3D structure of the mTOR protein along with its main domains for clarity. We also expanded the description of Figure 3 explaining the targeted domains for each generation of inhibitor (Line 414-422). We also reiterated and mentioned the figure in each of the subsections of mTOR inhibitors to ensure the reader is aware of their mode of action (Lines 335, 363, and 403). 

  1. Line 182: NLRC3 itself is a receptor; mention its ligands.

Response: We thank the reviewer for pointing this out. The reason we did not mention any ligands for NLRC3 is that it is one of the least studied NLRs and no research has shown its putative ligand(s). In fact, out of the approximately 20 NLRs in humans, only a few have been matched up with their putative ligands, but NLRC3 is not one of them. Additionally, it is not clear whether NLRs are able to recognize ligands directly through their leucine-rich repeats (LRRs) domain, which is known for protein-protein interactions. It remains unclear whether NLRC3 requires binding to a specific ligand, or if it is engaged in a ligand-independent manner to exerts its regulatory functions, including its role in mTOR signaling.

Reviewer 3 Report

Comments and Suggestions for Authors

In the present work, Marafie try to review the mTOR’s critical role in metabolic diseases, cancer, and the aging process. In addition, the latest findings on mTOR inhibitors and their clinical implications are reviewed, and the regulation of mTOR signaling could be used for therapeutic strategies. However, there are some questions that should be explained.

Major concerns

1. As a high impact factor Journal for a review paper, additional fine Figures should be included, which may be related to the mTOR regulation in metabolic diseases and mTOR signaling in cancer, as well as the differences between mTORC1 and mTORC2, or mTOR signaling and aging process.

2. As a review article, the latest progresses in this area should be included. However, some papers that are closely related to this article are not cited.

Glaviano A, Foo ASC, Lam HY, Yap KCH, Jacot W, Jones RH, Eng H, Nair MG, Makvandi P, Geoerger B, Kulke MH, Baird RD, Prabhu JS, Carbone D, Pecoraro C, Teh DBL, Sethi G, Cavalieri V, Lin KH, Javidi-Sharifi NR, Toska E, Davids MS, Brown JR, Diana P, Stebbing J, Fruman DA, Kumar AP. PI3K/AKT/mTOR signaling transduction pathway and targeted therapies in cancer. Mol Cancer. 2023;22(1):138.

Mafi S, Ahmadi E, Meehan E, Chiari C, Mansoori B, Sadeghi H, Milani S, Jafarinia M, Taeb S, Mafakheri Bashmagh B, Mansoorian SMA, Soltani-Zangbar MS, Wang K, Rostamzadeh D. The mTOR Signaling Pathway Interacts with the ER Stress Response and the Unfolded Protein Response in Cancer. Cancer Res. 2023;83(15):2450-2460.

3. Lines 83-142, ‘2.1. mTOR Regulation in Metabolic Diseases’ subsection, there are three paragraphs, different paragraph should focus different metabolic diseases, for example, diabetes, obesity and fatty liver.

4. Lines 144-230, ‘2.2. mTOR Signaling Promotes Cancer’ subsection, there are four paragraphs, different paragraph should focus different cancers, for example, breast cancer, prostate cancer, colorectal cancer and others.

Minor concerns

1. There are many ‘will’ in Abstract section and other section, which are not suitable.

2. English grammar and writing style are poor. There are too much wrong, which should be checked throughout the manuscript. For example,

Lines 32-33, ‘diseases [3-5] Dysregulation’; Line 51, ‘glucose uptake. [3, 10-12].’

3. Introduction section is too long, and should be refined.

4. Lines 105-128, too long paragraph. This paragraph should be refined.

5. ‘3. mTOR Signaling and Life Extension’ section may have several subsections.

6. Lines 224-248, too long paragraph. This paragraph should be refined.

7. ‘third-generation inhibitor of mTOR’ may be a subsection.

8. Figure 1, mTOR molecule may be in a 3-D condition.

9. Figure 2 should be explained in detail.

10. The format of reference is not suitable for this Journal.

Comments on the Quality of English Language

Extensive editing of English language required.

Author Response

Authors’ response to Reviewer 3:

We thank the reviewer for the constructive feedback and input. The points raised were helpful and strengthened the impact of the review. We addressed the points raised by the reviewer and modified the manuscript accordingly by highlighting the changes within the manuscript in yellow. We hope the adjusted manuscript will satisfy the key comments pointed out by the reviewer. 

Reviewer 3:

In the present work, Marafie try to review the mTOR’s critical role in metabolic diseases, cancer, and the aging process. In addition, the latest findings on mTOR inhibitors and their clinical implications are reviewed, and the regulation of mTOR signaling could be used for therapeutic strategies. However, there are some questions that should be explained.

Major concerns

  1. As a high impact factor Journal for a review paper, additional fine Figures should be included, which may be related to the mTOR regulation in metabolic diseases and mTOR signaling in cancer, as well as the differences between mTORC1 and mTORC2, or mTOR signaling and aging process.

Response: We thank the reviewer for the constructive feedback and for pointing this out. We agree that the manuscript lacked any visual aid and the need for addition figures is necessary. We addressed this point by including two additional figures, Figure 1 and Figure 2. Figure 1 as part of the introduction illustrates the two main complexes of the mTOR signaling pathway along with their components and their different cellular roles. Figure 2 is part of Chapter 2 that discusses the roles of mTORC1 and mTORC2 in regulating metabolic diseases (i.e. diabetes, obesity, and NAFLD) and cancers, as well as their roles in aging. Figure 1 can be found in Line 71, while Figure 2 in Line 284.

  1. As a review article, the latest progresses in this area should be included. However, some papers that are closely related to this article are not cited.

Response: We thank the reviewer for pointing this out and we agree that the mentioned articled should be cited to include the latest progress in mTOR signaling. The newly cited articles for Glaviano et al. 2023 and Mafi et al. 2023 are references 7 and 44, respectively.

  1. Lines 83-142, ‘2.1. mTOR Regulation in Metabolic Diseases’ subsection, there are three paragraphs, different paragraph should focus different metabolic diseases, for example, diabetes, obesity and fatty liver.

Response: We thank the reviewer for the comments and constructive feedback. We agree that having more focused paragraphs for the different metabolic diseases allows for a better overall read. Since the obesity and fatty liver sections were brief in comparison to the diabetes one, we restructured the subsection as kindly suggested by the reviewer into three main parts as follows: the first paragraph for diabetes, the second paragraph for obesity and fatty liver, and the final paragraph for neuronal metabolism. Below is the modified subsection:

Lines 111-171:

“Many studies have demonstrated the effect of mTOR dysregulation on metabolic diseases such as diabetes, obesity, and other cardiovascular diseases [23, 24]. In diabetes, mTOR has been shown to play a key role in affecting insulin resistance and sensitivity, glucose uptake, lipid metabolism, and ketone production [23]. Diabetes is characterized with increased glucose and lipid levels that consequently disrupt the individual’s metabolic profile leading to hyperglycemia and hyperlipidemia, and eventually, insulin resistance [25, 26]. One of the key targets downstream mTORC1 signaling is the insulin receptor substrate 1 (IRS-1) that prevents constitutive activation of the pathway [27]. IRS-1 hyperactivation due continuous mTORC1 activity has been linked to both type 1 and type 2 diabetes where glucose fails to translocate to its surface receptor resulting in increased glucose levels in the blood (Figure 2 bottom panel) [24]. Moreover, others have implicated a potential crosstalk between Akt and IRS-1 via mTORC2 under lipotoxic conditions in rat insulinoma cells lines. The study also demonstrated a reduction in glucose-stimulated insulin secretion (GSIS) under the same conditions [28]. A recent study by Brouwers et al. demonstrated a novel molecular mechanism of one of the most extensively studied proprotein convertase (PC), furin, in regulating glucose levels in β-cells [29]. PCs are proteolytic enzymes that are ubiquitously involved in many biological process and play crucial roles in regulating metabolic diseases [30, 31].  In their study, they showed that the absence of furin mediated mTORC1 activation leading to β-cell dysfunction (Figure 2 bottom panel) [29]. The crucial roles of PCs in diabetes and its complications have been highlighted by others emphasizing their importance for the development of future therapeutic interventions [32]. Moreover, mTORC1 hyperactivation has been shown to have a protective role in diabetes [33]. Ketogenesis involves the breakdown of fat in the liver into ketone bodies, which serves as an energy source for neighboring organs [34]. Uncontrolled ketogenesis causes increased levels of glucose leading to hyperglycemia [35]. Ursino and colleagues demonstrate such protective role of mTORC1 where its increased activity in the liver hampers the production of ketone bodies. During fasting, the expression peroxisome proliferator activated receptor (PPAR), the main transcriptional activator of ketone bodies, is increased. Inhibiting PPAR is essential to promote mTORC1 activity, and therefore, regulate ketogenesis [33]. As a result, mTORC1 has been considered as a promising therapeutic target for restraining diabetic ketogenesis (Figure 2 top panel). Gayatri et al. demonstrated another example of an interplay between mTORC1 and mTORC2 under diabetic conditions where glutamine played a key role in regulating both pathways. Under conditions of low glutamine levels, mesenchymal stromal cells (MSCs) exhibit reduced mTORC1 activity, which in turn promotes mTORC2 stabilization of Runt-related tran-scription factor 2 (RUNX2), a critical transcription factor involved in bone differentiation within MSCs. They also showed that high glucose conditions trigger mTORC1 hyper-activity suppressing mTORC2 in a glutamine-dependent manner. This highlights an important role for glutamine in controlling the nutrient switch between both mTORC1 and mTORC2 signaling, in addition to its importance in modulating the molecular cues involved in driving diabetes-induced bone adipogenesis [36].

Dysregulation of mTOR signaling has also been shown to play a role in the context of obesity. Elevated triglycerides (TGs) and a reduction in lipoprotein lipase (LPL) ex-pression, two hallmarks of obesity, have been linked with the upregulation of sodium coupled neutral amino acid transporter 2 (SNAT2) in an mTORC1-dependent manner [37]. Constitutive activation of mTORC1 has also been shown to exacerbate obesity due to the accumulation of hepatic lipid deposits leading to insulin resistance [38, 39]. mTOR signaling has also been associated with nonalcoholic fatty liver disease (NAFLD). Using a Mendelian randomization approach, single-nucleotide polymorphisms (SNPs) found in eukaryotic translation initiation factor 4E (eIF4E), a mTORC1 target, from Genome-Wide Association Studies (GWAS) demonstrated an effect on NAFLD. Increased plasma levels of eIF4E were associated with an increased risk of developing NAFLD. This emphasizes the importance of mTOR-related pathways in liver health (Figure 2 bottom panel) [6].

Other aspects of metabolic diseases have also been explored where mTOR has been shown to be central. In the context of neuronal energy consumption and synaptic plasticity, a crosstalk between mTOR and its upstream kinase AMPK was established. Using computational modeling, the dynamics of both AMPK and mTOR activity were investigated in response to the amino acid glutamate. Moreover, the interplay between insulin receptor and calcium signaling on AMPK and mTOR activation was fed into the computational model to predict their potential effect on insulin signaling and the metabolic consumption rate which provides a better understanding of neuronal metabolism [1].”

  1. Lines 144-230, ‘2.2. mTOR Signaling Promotes Cancer’ subsection, there are four paragraphs, different paragraph should focus different cancers, for example, breast cancer, prostate cancer, colorectal cancer and others.

Response: We thank the reviewer for highlighting this point. We agree that each paragraph should focus on different types of cancers. We addressed this point by restructuring the subsection into three main bodies as follows: the first paragraph for on overall view of mTOR signaling in cancer, the second paragraph focuses on both breast and prostate cancers, and the final paragraph focuses on colorectal cancer. Below is the modified subsection:

Lines 173-234:

“The mTOR signaling pathway is exploited by approximately 30% of cancers con-tributing to their development and progression due to its various roles that impact cell cycle, growth, survival, and metabolism. It also regulates nutrient utilization and energy production emphasizing its crucial metabolic role promoting aspects of cancer progression and invasiveness such as angiogenesis and metastases [23, 40]. Dysregulation of cancer-critical genes leads to mTOR hyperactivation, which in turn, increases translation of pro-oncogenic proteins that directly affect cellular processes like cell growth, migration, and de novo blood vessel formation [41]. For instance, the dysfunction of both mTORC1 targets eIF-4E and 4E-BP1 has been implicated to affect such processes. De-creased 4E-BP1 and increased eIF-4E levels directly contribute to mTORC1 hyperactivation leading to pro-oncogenic traits [23, 24, 42]. Moreover, increased rates of protein synthesis are also accompanied by mTORC1 hyperactivation further promoting cancer development [43]. In terms of tumor metabolism, mTOR’s role as a nutrient sensor is key due to its response to different nutrient cues such as glucose, amino acids, growth factors, and other stressors [23, 44]. Controlling nutrient utilization and energy pro-duction provide the metabolic adaptations necessary for cancer cell survival and proliferation. Autophagy, a process that involves recycling cytoplasmic components in response to nutrient scarcity and lack of energy, is another cellular mechanism controlled by mTOR which suppresses carcinogenesis [45]. The absence of autophagy has been reported to promote the development of cancer [46, 47]. mTORC1 phosphorylates one of autophagy’s key players UNC-5-like autophagy-activating kinase 1 (ULK1) preventing it from complexing with other autophagic components, thereby activating autophagy [48]. mTORC2, on the other hand, has been shown to indirectly regulate mTORC1 and suppress autophagy (Figure 2 bottom panel) [49].

mTOR has been reported to be dysregulated in approximately 70% of breast cancers [50]. Triple-negative breast cancer (TNBC), a form of breast cancer that is more aggressive and has a higher susceptibility for metastases, has been reported to have mutations in the PI3K/Akt/mTOR pathway in 25% of the cases [51, 52]. Such disruption serves as a key TNBC survival and resistance coping mechanism, making it a promising target for the treatment of TNBC. mTORC2 signaling also demonstrates cancer-promoting roles due to its regulation of Akt, glucose uptake, and apoptosis promoting pro-proliferative cellular functions [23]. Rictor, one of the main components of mTORC2, has been re-ported to be crucial for the progression of human prostate cancer cell lines [53]. Phosphatase and tensin homolog (PTEN) negatively phosphorylates Rictor leading to the dissociation of mTORC2, therefore, abolishing its activity [54]. This inhibitory mechanism is continuously exploited in both mouse and human cell line prostate cancer models where mTORC2 signaling is constitutively activated by PTEN-deficiency or loss which contributes to prostate cancer progression (Figure 2 bottom panel) [53].

Despite mTOR hyperactivity in many cancers, some cytoplasmic regulators prevent mTOR activity preventing tumor growth. For instance, NLRC3 (nucleotide-binding domain and leucine-rich repeats 3) has been shown to negatively regulate mTOR signaling [19]. NLRs are a family of cytoplasmic sensors that regulate a range of biological functions involved in immunity against infectious diseases [55] and are central regulators of intestinal homeostasis [56]. Studies have shown that the expression of NLRC3 is dramatically reduced in patients with colorectal cancer (CRC), implicating its potential role in cancer development [57]. Karki and colleagues further confirm the role of NLRC3 in promoting colorectal tumorigenesis in mice in a mTOR-dependent manner. They demonstrate that upon NLRC3 binding to its receptor, it suppresses mTOR activation preventing cellular proliferation by blocking the activation of Akt downstream of PI3K. They reveal a key mTOR inhibitory role for NLRC3 mediating protection against CRC (Figure 2 top panel) [19]. Another research aspect that has been shown to play important roles in human diseases including cancers is the study of gut microbiota. p53-induced intestinal onco-genesis has been shown to be modulated epigenetically by the gut microbiome composition. Specifically, CRC was influenced by dietary proteins that impacted the gut microbiota which is crucial for the development of effective cancer therapies [58, 59]. A recent study compared the effect of two different diets, namely a casein protein diet (CTL) and a free amino acid (FAA)-based diet on the progression of CRC and gut microbiota in mice. They found that FAA significantly attenuated CRC progression com-pared to the CTL diet in addition to an enrichment of beneficial gut bacteria. The attenuation was due to the downregulation of several cancer-associated pathways, including mTOR signaling [60]. This further emphasizes the vast role mTOR signaling plays in cancer metabolism and the importance of gut microbial composition in im-proving gut functions and preventing carcinogenic pathways.”

Minor concerns

  1. There are many ‘will’ in Abstract section and other section, which are not suitable.

Response: We thank the reviewer for pointing this out. We addressed this point and adjusted the manuscript accordingly.

  1. English grammar and writing style are poor. There are too much wrong, which should be checked throughout the manuscript. For example,

Lines 32-33, ‘diseases [3-5] Dysregulation’; Line 51, ‘glucose uptake. [3, 10-12].’

Response: We thank the reviewer for highlighting these points. We addressed this by going thoroughly through the manuscript and adjusted it accordingly. We also adjusted other grammar and writing mistakes in the manuscript which are highlighted in yellow.

  1. Introduction section is too long, and should be refined.

Response: We thank the reviewer for their comment and constructive feedback. We refined the grammar and writing style of the introduction. Kindly note that it is important to mention the role and components of mTOR signaling in detail as a prelude for the upcoming chapters. Since it is part of a vast network of signaling pathways, it is crucial to include some detail about both mTOR’s upstream and downstream effectors as they play key roles in the diseases discussed in this review. It also helps understand what part of mTOR signaling is targeted by the different generations of inhibitors which are discussed later in the manuscript. However, we agree with the reviewer on how the introduction might feel lengthy, so we addressed this by including an additional figure (i.e. Figure 1) that translate the details mentioned which visually helps in breaking down the lengthiness of the introduction.

  1. Lines 105-128, too long paragraph. This paragraph should be refined.

Response: We thank the reviewer for the comments. We addressed this point as part of the restructuring of this subsection as kindly suggested by the reviewer in the “Major concerns” section.

  1. ‘3. mTOR Signaling and Life Extension’ section may have several subsections.

Response: We thank the reviewer for the comment and agree that this section would read better by breaking it into subsections. We addressed this by structuring the content into three subsections: the first talking about the evolutionary conserved role of mTOR in the aging process, the second for the role of autophagy in aging, and the final section for the role of mitochondrial proteins in regulating life span. Below is the modified content in detail:

Lines 235--322:

3. mTOR Signaling and Life Extension

3.1. The Role of mTOR Signaling in Aging is Evolutionary Conserved

The role of mTOR signaling with regards to aging has been highlighted in different studies and is shown to be conserved from yeast to mammals [23, 61]. Initial studies performed on C. elegans demonstrated the role of mTOR and Raptor orthologs on the longevity and the extension of life span in worms. They showed that decreased ex-pression both their mTOR and Raptor orthologs had a direct effect on life span extension [62, 63]. Similar studies performed on other organisms like budding yeast [64], Drosophila [65], and mouse models resulted in consistent findings [66, 67]. Nutrient deprivation was suggested as a means of lowering the organism’s metabolic rate leading to life span extension. Indeed, limiting nutrients led to a reduction in mRNA translation driven by mTORC1, and eventually, lowered oxidative stress and prevented the accumulation of toxic metabolites causing an increase in life expectancy [23]. Others re-ported similar findings in mammals further confirming the link between lower mTORC1/S6K1 activity and increased life span (Figure 2 top panel) [68]. Additionally, the mTOR inhibitor rapamycin has also been shown to have a role in life span extension across different organisms making it the only known pharmaceutical drug to directly regulate aging [69-72].

3.2. Autophagy Promotes Life Span Extension in a mTOR-dependent Manner

One explanation that correlates mTORC1 activity to life span extension is the role of autophagy in mTOR signaling. Inhibition of mTORC1 promotes autophagy, which in turn cleanses unwanted cytosolic proteins and reduces the accumulation of toxic metabolites leading to a reduction in cellular stress and extension of life span [23]. To date, it has been established that mTOR signaling inhibits autophagy, and autophagy declines with aging [73, 74]. This has been further highlighted in recent studies investigating senescent cells in the context of aging. One of the hallmarks of aging is the accumulation of senescent cells which are commonly present in many age-related diseases and cancers [75]. Recently, studies have demonstrated increased expression of programmed death-ligand 1 (PD-L1) protein in senescent cells. Upregulation of PD-L1 protects senescent cells from clearance by the PD-1 checkpoint receptor, and mTOR signaling has been shown to be one of the key stimulators of PD-L1 [76]. Thus, the PD-1/PD-L1 checkpoint is crucial for promoting the accumulation of senescent cells and slower aging (Figure 2 top panel). Moreover, there has been evidence demonstrating that mTOR-mediated inhibition of autophagy stimulates PD-L1 expression [77-79]. Others also showed similar findings where autophagy inhibition, by either pharmacological inhibitors or knockout of autophagic genes, increased PD-L1 expression in mouse and human cancer cell models. It is known that the PD-L1 possesses functions other than the cell-intrinsic ones [80, 81]. One study demonstrated that PD-L1 acts as an upstream regulator of mTOR activating the Akt-mTOR axis promoting the proliferation of human ovarian cancer cells [82]. Interestingly, others have reported PD-L1’s indirect mediation of mTOR signaling by directly preventing the autophagy flux in tumor cells [83]. This highlights the bidirectional role PD-L1 plays in regulating mTOR and autophagy that seems to be cell-specific in certain cases. Taken together, these findings suggest PDL-1as a potential candidate for anti-aging therapies and interventions.

3.3. Mitochondrial Proteins Regulate Life Span via mTOR Signaling

Another hallmark of aging is the process of controlling the coordination between mitochondrial and nuclear activities that are essential for overall cellular respiration. Disruption of such coordination leads to mitochondrial dysfunction, affecting overall aging [75, 84]. Signaling pathways drive the crosstalk between the mitochondria and the nucleus, which in turn affects nuclear gene expression to maintain mitochondrial homeostasis [85, 86]. Clk-1 is one of the key proteins that has been shown to play a role in such communication, affecting cellular respiration and longevity [87]. A study conducted by Monaghan et al. uncovered a direct nuclear role for Clk-1 that regulates life span which is conserved from C. elegans to humans [88]. It was later revealed for the first time that Clk-1 could be regulated by mTOR via the AMPK-mTOR axis [89]. Thus, these studies emphasize the crucial role mTOR signaling plays in regulating life span by communicating and linking with different biological processes by sensing various cellular cues.”

  1. Lines 224-248, too long paragraph. This paragraph should be refined.

Response: We thank the reviewer for the feedback. We agree with the reviewer and the paragraph has been refined. We removed some redundant studies that had similar findings to what has been mentioned already. However, kindly note we think it is important to mention some of the comparison studies in this section as it highlights the bidirectional role PD-L1 plays in regulating mTOR signaling. Below is the refined paragraph:

Line 256-276:

“To date, it has been established that mTOR signaling inhibits autophagy, and autophagy declines with aging [73, 74]. This has been further highlighted in recent studies investigating senescent cells in the context of aging. One of the hallmarks of aging is the accumulation of senescent cells which are commonly present in many age-related diseases and cancers [75]. Recently, studies have demonstrated increased expression of programmed death-ligand 1 (PD-L1) protein in senescent cells. Upregulation of PD-L1 protects senescent cells from clearance by the PD-1 checkpoint receptor, and mTOR signaling has been shown to be one of the key stimulators of PD-L1 [76]. Thus, the PD-1/PD-L1 checkpoint is crucial for promoting the accumulation of senescent cells and slower aging (Figure 2 top panel). Moreover, there has been evidence demonstrating that mTOR-mediated inhibition of autophagy stimulates PD-L1 expression [77-79]. Others also showed similar findings where autophagy inhibition, by either pharmacological inhibitors or knockout of autophagic genes, increased PD-L1 expression in mouse and human cancer cell models. It is known that the PD-L1 possesses functions other than the cell-intrinsic ones [80, 81]. One study demonstrated that PD-L1 acts as an upstream regulator of mTOR activating the Akt-mTOR axis promoting the proliferation of human ovarian cancer cells [82]. Interestingly, others have reported PD-L1’s indirect mediation of mTOR signaling by directly preventing the autophagy flux in tumor cells [83]. This highlights the bidirectional role PD-L1 plays in regulating mTOR and autophagy that seems to be cell-specific in certain cases. Taken together, these findings suggest PDL-1as a potential candidate for anti-aging therapies and interventions.”

  1. ‘third-generation inhibitor of mTOR’ may be a subsection.

Response: We thank the reviewer for pointing this out and agree that having a separate subsection for the third-generation inhibitors enhances the overall section. We restructured the content to address the first three generations of inhibitors sequentially, and then closing the section with other mTOR inhibitors (i.e. metformin and Tau). The new subsections can be found in Lines 401 and 423.

  1. Figure 1, mTOR molecule may be in a 3-D condition.

Response: We thank the reviewer for the suggestion. We addressed this by adding an extra panel to Figure 3 (formerly Figure 1) which illustrates a 3-D structure of mTOR with its four domains generated using Alpha Fold 2.0. We used the same color codes of the mTOR domains on both the schematic and 3-D panels of the figure which can be found in Line 413.

  1. Figure 2 should be explained in detail.

Response: We thank the reviewer for mentioning this. We agree that Figure 4 (formerly Figure 2) was not explained clearly and have addressed this point. We expanded the figure legend to explain in detail the different physiological states in response to mTOR activity levels. Below are the changes:

Line 472-484:

“Figure 4. A summary of the central role mTOR signaling plays in regulating various cellular processes and its association with metabolic diseases, cancers, and aging. The left panel (green) illustrates the physiological state where mTOR signaling is reduced. Under these conditions, cells are characterized by having a balanced metabolism (i.e. normal glucose and lipid levels), being sensitive to insulin, a reduction in stress and toxicity, and having a balanced cell cycle. As a result, lower mTOR activity leads to an improvement in overall health, controlled cell growth, and eventually, a prolonged life span. Upon mTOR hyperactivity (right panel, red), glucose and lipid levels increase, cells undergo insulin resistance, stress and toxicity levels in-crease, and the cell cycle is disrupted promoting uncontrolled cell growth. Such disruptions in mTOR signaling directly contribute to the progression of metabolic diseases (e.g. diabetes and obesity), cancers, and consequently, reducing life span. The right panel also shows the impact of mTOR inhibitors, like metformin and rapamycin, and their effects in mitigating diseases progression and life span extension by reversing the drastic effects caused by mTOR hyperactivity (Created with BioRender.com).”

  1. The format of reference is not suitable for this Journal.

Response: We thank the reviewer for the feedback. Kindly note we followed the International Journal of Molecular Sciences’ guidelines for referencing, and they indicate that the references can be in any style as long as they are consistent. We also looked at previous publication in IJMS to ensure we followed the proper guidelines. EndNote was used as the bibliography software in this review. Below is the quote from the IJMS “Instructions for Authors” web page:

“Your references may be in any style, provided that you use the consistent formatting throughout. It is essential to include author(s) name(s), journal or book title, article or chapter title (where required), year of publication, volume and issue (where appropriate) and pagination. DOI numbers (Digital Object Identifier) are not mandatory but highly encouraged. The bibliography software package EndNote, Zotero, Mendeley, Reference Manager are recommended.”

Round 2

Reviewer 3 Report

Comments and Suggestions for Authors

Thanks for author’s responses. However, the format of reference should be referred the published paper in this IJMS.

Comments on the Quality of English Language

Minor editing of English language required.